# CRISPR/Cas9-Mediated Targeted Mutagenesis of *BnaCOL9* Advances the Flowering Time of *Brassica napus* L.

**DOI:** 10.3390/ijms232314944

**Published:** 2022-11-29

**Authors:** Jian Guo, Lei Zeng, Hui Chen, Chaozhi Ma, Jinxing Tu, Jinxiong Shen, Jing Wen, Tingdong Fu, Bin Yi

**Affiliations:** National Key Laboratory of Crop Genetic Improvement, Hubei Hongshan Laboratory, College of Plant Science and Technology, National Center of Rapeseed Improvement in Wuhan, Huazhong Agricultural University, Wuhan 430070, China

**Keywords:** *BnaCOL9*, *Brassica napus*, CRISPR/Cas9, flowering time

## Abstract

Rapeseed (*Brassica napus* L.) is one of the most important oil crops in the world. The planting area and output of rapeseed are affected by the flowering time, which is a critical agronomic feature. *COL9* controls growth and development in many different plant species as a member of the zinc finger transcription factor family. However, *BnaCOL9* in rapeseed has not been documented. The aim of this study was to apply CRISPR/Cas9 technology to create an early-flowering germplasm resource to provide useful material for improving the early-maturing breeding of rapeseed. We identified four *COL9* homologs in rapeseed that were distributed on chromosomes A05, C05, A03, and C03. We successfully created quadruple *BnaCOL9* mutations in rapeseed using the CRISPR/Cas9 platform. The quadruple mutants of *BnaCOL9* flowered earlier than the wild-type. On the other hand, the flowering time of the *BnaCOL9* overexpression lines was delayed. An analysis of the expression patterns revealed that these genes were substantially expressed in the leaves and flowers. A subcellular localization experiment demonstrated that *BnaCOL9* was in the nucleus. Furthermore, we discovered that two key flowering-related genes, *BnaCO* and *BnaFT*, were highly elevated in the *BnaCOL9* mutants, but dramatically downregulated in the *BnaCOL9* overexpression lines. Our findings demonstrate that *BnaCOL9* is a significant flowering inhibitor in rapeseed and may be employed as a crucial gene for early-maturing breeding.

## 1. Introduction

Rapeseed is an important oil and economic crop worldwide. *B. napus* (AACC, 2n = 38) was generated 7500 years ago from a natural hybrid of *Brassica rapa* (AA, 2n = 20) and *Brassica oleracea* (CC, 2n = 18) and subsequently underwent chromosomal doubling to form an allotetraploid plant [1]. Rapeseed is classified into three ecotypes based on the environment required for flowering: winter, semi-winter, and spring. Depending on local circumstances, specific ecotypes can be planted in various regions. Therefore, the optimal flowering time for rapeseed planting is critical as it influences not only the production but also other crop rotations [2]. As a result, the flowering time is an essential feature for breeders throughout the rapeseed breeding process.

The flowering regulation network in Arabidopsis has been extensively researched; it primarily contains the photoperiod, vernalization, temperature, autonomy, age, gibberellin (GA), and sugar metabolism pathways [3,4]. Among these pathways, the flowering pathway contains several important genes known as integrators; for example, *SOC1* (*SUPPRESSOR OF OVEREXPRESSION OF CONSTANS1*), *CO* (*CONSTANS*), and *FT* (*FLOWERING LOCUS T*). *SOC1* encodes a MADS-box transcription factor that integrates flowering signals from the photoperiod, temperature, hormone, and age pathways [5]. *CO* was the first gene identified in the BBX family and regulates the flowering time via the photoperiod pathway [6]. *FT* encodes a phosphatidylethanolamine-binding protein, which is a small mobile protein. This gene is a major integrator in the control of the flowering pathway [7].

*COL9* (*CONSTANS-like 9*) and *CO* belong to the BBX gene family and they encode zinc finger transcription factors [8,9]. There are 32 known *B-BOX* genes in Arabidopsis, which may be classified into five groups (I–V) based on amino acid variations in the *B-BOX* motif and CCT domain [9]. *COL9* can suppress the expression of *FT* and *CO*, delaying the flowering time in Arabidopsis [10]. Other *B-BOX* genes that have been linked to flowering time regulation in Arabidopsis include *BBX4*/*COL3*, *BBX6*, *BBX10*, *BBX19*, *BBX24*, *BBX30*, *BBX31*, and *BBX32* [11,12,13,14,15]. *BBX32* co-regulates flowering through its interaction with *BBX4/COL3*, which may be an additional regulatory mechanism affecting reproductive development in Arabidopsis [15]. In rice, *OsCOL3* negatively modulates the expression of *Hd3a* and *TFL* under short-day conditions to regulate the flowering time [16]. *OsCOL9* can delay the flowering time by inhibiting the *Ehd1* pathway, which is consistent with the results of the research on Arabidopsis [17]. Recent research has demonstrated that the chrysanthemum gene *CmBBX8* controls the flowering time by activating the *CmFTL* promoter [18].

According to the research, rapeseed and Arabidopsis are both members of the Brassicaceae family and may share a common ancestor [19,20]. Arabidopsis and rapeseed homologous genes share a high sequence similarity and may also have similar functions [21]. Until recently, there have been few reports on the flowering time of rapeseed. Previous research on the rapeseed flowering time has mostly focused on QTL linkage mapping and several QTL links to the flowering time have been discovered, primarily on chromosomes A02, A09, A10, C02, and C03 [22,23,24]. Genome-wide association studies (GWASs) have recently been used for the genetic dissection of complex features in plants such as rice [25], maize [26], and rapeseed [27,28,29]. A substantial number of single-nucleotide polymorphism (SNP) sites linked to the flowering time in rapeseed have been identified using GWAS analyses. A GWAS analysis of 523 natural populations under 8 distinct environmental conditions revealed that 41 SNPs were strongly associated with the flowering time [28]. Wu et al. found SNPs in the promoter regions of the *FT* and *FLC* (*FLOWERING LOCUS C*) orthologs correlating to distinct rapeseed ecotypes by resequencing 991 rapeseed accessions globally and conducting a GWAS analysis [29]. These studies may provide important molecular markers for accelerating *Brassica napus* breeding.

This study is novel because it used the CRISPR/Cas9 system to edit four *BnaCOL9* homologs in rapeseed and generated a quadruple mutant of *BnaCOL9*, which has not been previously reported on to the best of our knowledge. Our results indicated that the application of CRISPR/Cas9 technology to create early-flowering germplasm resources provided useful materials to improve the early-maturing breeding of rapeseed.

## 2. Results

### 2.1. Isolation of BnaCOL9 from Brassica napus

Using the protein sequence of Arabidopsis *AtCOL9*, four *Brassica napus* orthologs of COL9—*Bna*A0529980D, *Bna*C0544310D, *Bna*A0330130D, and *Bna*C0335440D—were identified and named *BnaA05COL9*, *BnaC05COL9*, *BnaA03COL9*, and *BnaC03COL9*, respectively. The amino acid sequence identities with Arabidopsis *AtCOL9* were 79.63%, 79.85%, 72.95%, and 72.24%, respectively. These results suggested that the *BnaCOL9* homologs could have functions similar to those of *AtCOL9*. A phylogenetic analysis showed that the *BnaA05COL9* and *BnaA03COL9* proteins were most closely related to the homologs of *B. rapa*. The *BnaC05COL9* and *BnaC03COL9* proteins were most closely related to their homologs in *B. oleracea* (Figure 1A). A MEME analysis showed that *BnaA05COL9*, *Bna*A03*COL9*, *BnaC05COL9*, and *BnaC03COL9* contained ten conserved motifs (Figure 1B). In addition, the amino acid sequences of the four *BnaCOL9* homologs shared an 83.29% identity. The *B-BOX* domain was located at the N-terminal and the CCT domain was located at the C-terminal (Figure 1C). Therefore, these four *Bna*COL9 proteins could have redundant functions.

### 2.2. Knockout of BnaCOL9 Shows an Early-Flowering Phenotype

*Brassica napus* is an allotetraploid species, which suggests that it has at least two copies of each gene. *Brassica napus* has four copies of *BnaCOL9* and the biological functions may be redundant. To analyze the biological function of *BnaCOL9*, we designed two sgRNAs to knock out *BnaA05COL9*, *BnaC05COL9*, *BnaA03COL9*, and *BnaC03COL9* in rapeseed. sgRNA1 (sg1) was designed on the second exon of *BnaA05COL9*, *BnaC05COL9*, and *BnaC03COL9*; it was on the third exon of *BnaA03COL9*. sgRNA2 (sg2) was used to target the fourth exon of *BnaA05COL9*, *BnaC05COL9*, and *BnaC03COL9* as well as the fifth exon of *BnaA03COL9* (Appendix A). In total, we obtained 40 T0 regenerated transgenic plants and 38 of them were positive. The primers (Cas9-F and Cas9-R) used to detect the positive plants are listed in Appendix A. The percentage of positive plants was 95% (Appendix A). Among the T1 regeneration lines, three independent mutant lines (T1-*cr6*, T1-*cr7,* and T1-*cr9*) in which four copies of *BnaCOL9* were simultaneously edited were selected for further research (Appendix A). In the T2 generation, we used Hi-TOM sequencing to confirm the presence of targeted mutations on the target site (Appendix A); most of the edit types were single-base insertions. The types of the base deletion mutations, including single-base, four-base, and five-base deletions, were also detected (Figure 2A). The flowering time was assessed and compared with that of the wild-type. Notably, the mutant lines showed earlier flowering times than the wild-type plants (Figure 2B). The average flowering time of the mutant lines, including T2-*cr6*, T2-*cr7*, and T2-*cr9*, was 34.67 ± 1.97 d, 35.16 ± 1.94 d, and 35.33 ± 1.94 d, respectively. In comparison, the average flowering time of the wild-type was 43 ± 1.26 d (Figure 2C). This result indicated that the knockout of *BnaCOL9* resulted in an early flowering.

### 2.3. Overexpression of BnaCOL9 Delays the Flowering Time

To confirm that *BnaCOL9* was involved in regulating the flowering time in rapeseed, we generated *BnaCOL9*-mCherry OE lines in a Westar genetic background and assessed their flowering time. The *BnaCOL9* overexpression lines showed a late flowering time compared with the wild-type plants (Figure 3A). The average flowering time of the overexpression lines OE-1, OE-3, and OE-4 was 52 ± 2.28 d, 51.5 ± 1.60 d, and 51.8 ± 1.95 d, respectively, which was significantly later than that of the wild-type (Figure 3B). The qRT-PCR results showed that the expression of *BnaA05COL9* was significantly increased in the overexpression lines (Figure 3C). Collectively, these results suggested that *BnaCOL9* regulates flowering in rapeseed.

### 2.4. BnaCOL9 Expression Pattern Analysis and Subcellular Localization

To study the expression pattern of *BnaCOL9* in rapeseed, a qRT-PCR analysis was performed using RNA extracted from the roots, leaves, stems, flowers, flower buds, and siliques. The results showed that *BnaA05COL9* and *BnaC05COL9* had similar expression patterns, which were highest in the leaves, followed by the roots. The transcript of *BnaA05COL9* was not detected in the siliques. The main expression tissues differed between *BnaA03COL9* and *BnaC03COL9*. *BnaA03COL9* was mainly expressed in the flowers. *BnaC03COL9* was mainly expressed in the leaves, followed by the flowers (Figure 4A). To further examine the expression pattern of *BnaA05COL9*, a construct harboring the GUS gene driven by the promoter of *BnaA05COL9* derived from Westar was transformed into rapeseed. Strong GUS signals were detected in the leaves and only weak GUS signals were detected in the roots, flowers, and buds (Figure 4B–E). However, we observed no GUS signals in the siliques or stems. The GUS expression pattern was consistent with the results of the qRT-PCR analysis.

The *BnaCOL9* protein was transiently transformed into tobacco plants through an Agrobacterium-mediated technique to determine its subcellular localization. The four BnaCOL9-GFP proteins were all localized in the nucleus (Figure 5), indicating that *BnaCOL9* is a nuclear protein. This was consistent with previous reports of it acting as a transcription factor to regulate various biological processes.

### 2.5. Transcription of Flowering-Related Genes Is Regulated by BnaCOL9

*COL9* has been found to directly regulate the flowering-related genes in Arabidopsis plants. The current study used qRT-PCR to detect the floral-related gene expression, including that of *BnaFT*, *BnaCO*, and *BnaSOC1*. The expression levels of these genes were measured in the leaves of the 28-day-old seedlings of the wild-type, *BnaCOL9* mutant, and *BnaCOL9* overexpression lines (Figure 6). In the quadruple mutant, the expression levels of *BnaFT* and *BnaCO* were significantly higher than those in the wild-type. However, the expression of *BnaSOC1* did not change. In contrast, the expression levels of *BnaFT* and *BnaCO* were significantly reduced in the overexpression lines. The expression of *BnaSOC1* remained unchanged. This was consistent with previous reports that *COL9* controls the flowering time by inhibiting the *FT* and *CO* expression in Arabidopsis. In conclusion, *BnaCOL9* might regulate the flowering time by inhibiting the expression of the *BnaFT* and *BnaCO* genes.

## 3. Discussion

The CRISPR/Cas9 gene editing system plays a vital role in studying gene functions and improving crop agronomic traits. Since the first successful application of this technology in plants [30], it has been applied to rice [31], wheat [32], soybeans [33], maize [34], and *Brassica napus* [35]. For example, a homozygous *Brassica napus Bnasvp* mutant was obtained using the CRISPR/Cas9 system and resulted in the stable transformation of *Brassica napu*s. It was found that the *Bnasvp* mutant showed an earlier flowering time under two environmental conditions (summer and winter growing seasons) [36]. *Brassica napus* with multilocular siliques was obtained by editing the *BnaCLV* gene using the CRISPR/Cas9 system [35]. In the present study, two synthetic single-guide RNAs (sg1 and sg2) were created with the intention of editing four conserved areas in the *BnaCOL9* genes. Through gene editing assays, we found that most of the mutations caused by sg1 and sg2 to the four copies of *BnaCOL9* were single-base insertions (Figure 2A). This could lead to a frameshift mutation in the *BnaCOL9* protein.

The flowering time is a complex biological process for plants, and research has revealed that approximately 300 genes in Arabidopsis are involved in the flowering regulation [37]. Although an overexpression of *COL9* in Arabidopsis has been shown to delay flowering [10], to the best of our knowledge, no reports have been published on whether *COL9* can affect the flowering of *Brassica napus*. We obtained mutants of *BnaCOL9* through gene editing and demonstrated that *BnaCOL9* could advance the flowering time (Figure 2), but an overexpression of *BnaCOL9* could delay the flowering time (Figure 3). By detecting the expression levels of the flowering-related genes in the overexpression and knockout mutants, the expression levels of *CO* and *FT* in the knockout mutants were found to be significantly increased whereas those in the overexpression lines were significantly decreased (Figure 6). This indicated that *BnaCOL9* might influence the flowering time through the regulation of *CO* and *FT* directly or indirectly in rapeseed. At present, however, we have no direct evidence to explain how *BnaCOL9* regulates flowering in *Brassica napus*; this will be the subject of our future work.

An important objective in breeding is to reduce the length of the vegetative stage of crops without significantly reducing the crop yields. Our study showed that certain CRISPR/Cas9 mutations may significantly enhance the agronomic features of *Brassica napus*. Mutants with earlier flowering times could be created by altering multiple copies of *BnaCOL9*. By modifying the genes associated with the flowering time, we could produce mutants with various flowering times, giving us more possibilities for early-maturing breeding in *Brassica napus*.

## 4. Materials and Methods

### 4.1. Plant Materials

For the transformation experiments, a spring-type rapeseed cultivar (Westar) was used as the receptor material. We obtained seeds from the National Rapeseed Engineering and Technology Research Center in Wuhan, China. The transgenic and wild-type plants were planted in a greenhouse (16 h light/8 h dark at 22 °C).

### 4.2. Construction of Phylogenetic Tree and Sequence Alignment

The Arabidopsis *AtCOL9* amino acid sequence was submitted to the Darmor-bzh database (https://www.genoscope.cns.fr/brassicanapus/, last access date: 24 November 2022) to search for *BnaCOL9* homologs in the rapeseed genome. The following species had full-length protein sequences of *BnaCOL9* and its homologs: *B. rapa*, *Brassica oleracea*, *Ziziphus jujuba*, *Rosa chinensis*, *Corchorus olitorius*, *Carica papaya*, *Theobroma cacao*, *Glycine max*, *Rhamnella rubrinervis*, and Arabidopsis. We retrieved these sequences from the NCBI database (https://www.ncbi.nlm.nih.gov/, last access date: 24 November 2022) and constructed a phylogenetic tree to study their evolutionary relationships. MEGA Ⅹ was used to build adjacency unrooted trees [38]. Protein-conserved motifs from *COL9* homologs were identified using MEME [39] and TBtools software (version 1.0987650) [40]. An alignment of multiple sequences was performed using DNAMAN software (version 6.0 Lynnon Biosoft, San Ramon, CA, USA).

### 4.3. Construction of a CRISPR/Cas9 Vector

We designed two sgRNAs using the CRISPR-P2.0 tool (http://crispr.hzau.edu.cn/CRISPR2/, last access date: 24 November 2022) to generate the *BnaCOL9* quadruple mutants. A CRISPR/Cas9 binary vector set was provided by Prof. Chen Qijun (China Agricultural University) that included PCBC-D1T2 (with a chloramphenicol resistance) and PKSE401 (with a kanamycin resistance) vectors. The *Zea mays* Cas9 protein was driven by the double 35S promoter and the two sgRNAs were driven by Arabidopsis promoters U6-26 and U6-29. They were fused to the T-DNA region of the PKSE401 vector. The vector was constructed as previously described [41]. We performed the experiments following the improved Agrobacterium-mediated genetic transformation method of Dai et al. [42]. The transformation steps were as follows: 1. First, we sowed seeds of sterilized rapeseed on an M0 medium and cultured them in a dark room for 7 days. The hypocotyl was picked out (the cotyledons and roots were removed). 2. We prepared the infested Agrobacterium bacterium solution. 3. We prepared the co-culture. 4. We performed the callus induction. The explants were cultured on the medium of M2 for 21 days. 5. We performed the shoot differentiation. The explants on M2 were transferred to the medium of M3 for 14 days; after that, the explants were transferred to the new medium of M3 every 14 days until the shoots came out. 6. We checked the root system for the initiation and transplantation of the seedlings. The shoots were cut from the junctions of the explants and transplanted to a clear square box of M4 for the culture. Usually, it took 2–3 weeks for the roots to grow.

### 4.4. Transgenic Plant Mutation Site Identification

We used a high-throughput sequencing platform to detect the mutant sites in the transgenic plants and submitted the sequencing results to the Hi-TOM decoding website to visualize the mutant sites [43]. The specific steps were as follows: 1. Target amplification using specific primers (first round of PCR, Appendix A). 2. Differential labeling of each plant with universal primers (second round of PCR). 3. Aliquots of the second round PCR products were mixed and subjected to next-generation sequencing (Annoroad Gene Technology, Beijing, China). 4. The sequencing results were decoded using an online decoding tool (http://www.hi-tom.net/hi-tom/, last access date: 24 November 2022) to show the mutation status of the target site.

### 4.5. RNA Extraction and qRT-PCR

RNA was extracted from the roots, stems, leaves, flowers, buds, and siliques of Westar to measure the expression of *BnaCOL9*. The total RNA from the transgene plants was extracted to detect the transcript levels of *BnaCOL9*, *BnaFT*, *BnaCO*, and *BnaSOC1*. Total RNA was extracted from the tissue samples using an RNA prep plant kit (Promega, Shanghai, China) containing a DNase I treatment reagent after harvesting and freezing in liquid nitrogen. First-strand cDNA was reverse-transcribed using a Transcript RT Kit (R223, Vazyme, Nanjing, China). For the qRT-PCR analysis, three biological replicates were used. According to the manufacturer’s protocol, the relative gene expression was determined using SYBR qPCR Master Mix (Q311, Vazyme) with a CFX384 touch real-time PCR detection system (Bio-Rad, Hercules, CA, USA). The 2^−ΔΔCt^ method was used to calculate the relative expression levels [44]. The mean values for each measured parameter were compared using two-tailed, two-sample Student’s *t*-tests in GraphPad Prism (version 8.0, GraphPad Software, San Diego, CA, USA) as appropriate.

### 4.6. Subcellular Localization

The coding sequences of *BnaAO5COL9, BnaCO5COL9, BnaAO3COL9*, and *BnaC03COL9* were amplified without the termination codon using specific primers (Appendix A). To generate the C-terminal GFP fusion products, the sequences were inserted into the transient expression vector PH7Lic6.0-35S-GFP. The constructed fusion plasmid of green fluorescent protein was transferred into Agrobacterium GV3101. The transformed cells were harvested and resuspended in 10 mM MES-KOH (pH 5.6) containing 10 mM MgCl_2_ and 150 mM acetosyringone and then mixed. The final optical density (OD600) at 600 nm was 0.8. After incubation for 2 h at 22 °C, the Agrobacterium suspension was injected into the expanded leaves of 4-week-old tobacco plants (*Nicotiana benthamiana*). The fluorescence signals in the tobacco leaves were imaged using a confocal laser scanning system (Leica SP8 DLS, Germany) [45].

### 4.7. GUS Assay

A 1.6 kb upstream sequence from the translation start site of *BnaAO5COL9* was cloned from Damor- bzh. This sequence was cloned into the pCAMBIA2300 binary vector. As previously described, the p*BnaA05COL9*-GUS fusion construct was introduced into Westar via an Agrobacterium-mediated transformation [42]. Positive transgenic plants were identified using the primers M13-47 and GUS-R (Appendix A). The GUS activity was determined by staining different tissues from positive transgenic plants in an X-GLU solution overnight at 37 °C. The tissues were then cleared with 75% (*v*/*v*) ethanol until the pigments were completely removed.

## Figures and Tables

**Figure 1 ijms-23-14944-f001:**
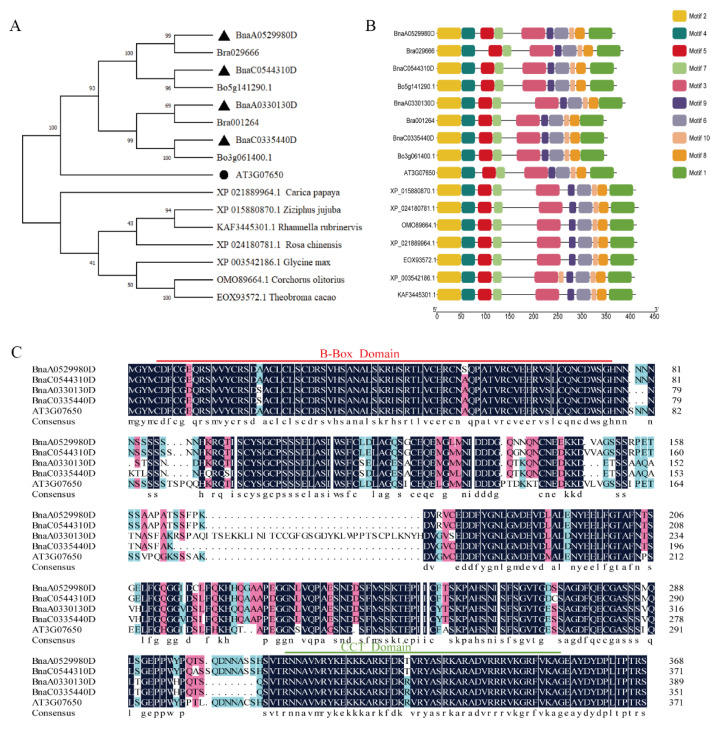
Sequence alignment of *BnaCOL9* and its phylogenetic tree: (**A**) phylogenetic tree of *Bna*COL9 and its protein homologs from *Brassica rapa*, *Brassica oleracea*, *Ziziphus jujuba*, *Rosa chinensis*, *Corchorus olitorius*, *Carica papaya*, *Theobroma cacao*, *Glycine max*, *Rhamnella rubrinervis*, and *Arabidopsis thaliana* (triangles represent four copies of *BnaCOL9*; circles represent *At3G07650*); (**B**) conserved motifs of *BnaCOL9* and its homologs (different motifs are represented by various colored boxes); (**C**) alignment of *AtCOL9* (*AT3G07650*) and *BnaCOL9* amino acid sequences (black shading indicates residues that are identical or similar, numbers indicate amino acid positions, the red line represents the *B-Box* domain region, and the green line represents the CCT domain).

**Figure 2 ijms-23-14944-f002:**
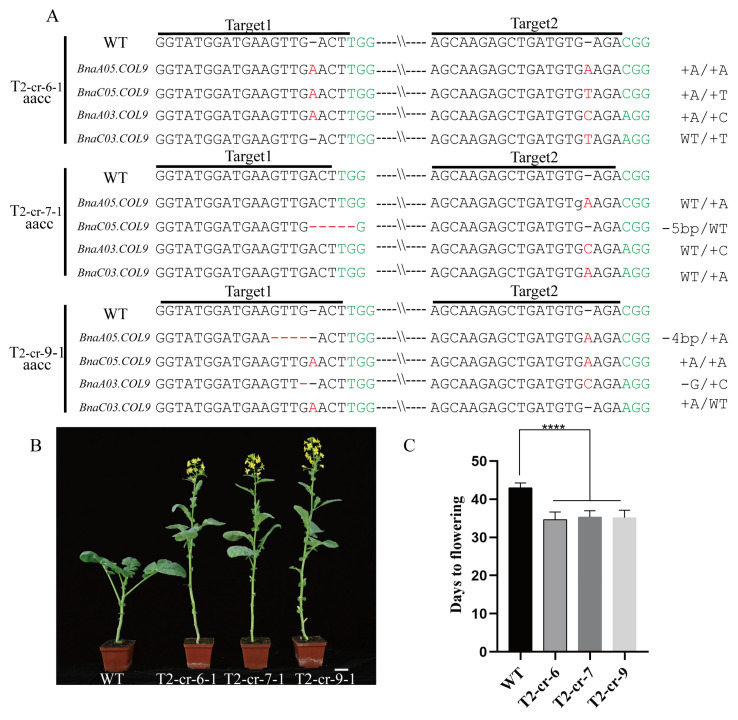
Quadruple mutants exhibited an early flowering: (**A**) sequence analysis of target sites in knockout lines of three mutants (T2-cr-6-1, T2-cr-7-1, and T2-cr-9-1) with wild-type (WT) sequences shown at the top (the target site is below the black line; the green font represents the PAM sequence, red fonts indicate insertions, and red short dashed lines indicate deletions); (**B**) flowering phenotypes of wild-type and T_2_ quadruple mutants (T2-cr-6-1, T2-cr-7-1, and T2-cr-9-1; scale bar = 5 cm); (**C**) flowering time statistics of wild-type and quadruple mutants (**** represents *p*  <  0.0001; Student’s *t*-test, *n*  =  6).

**Figure 3 ijms-23-14944-f003:**
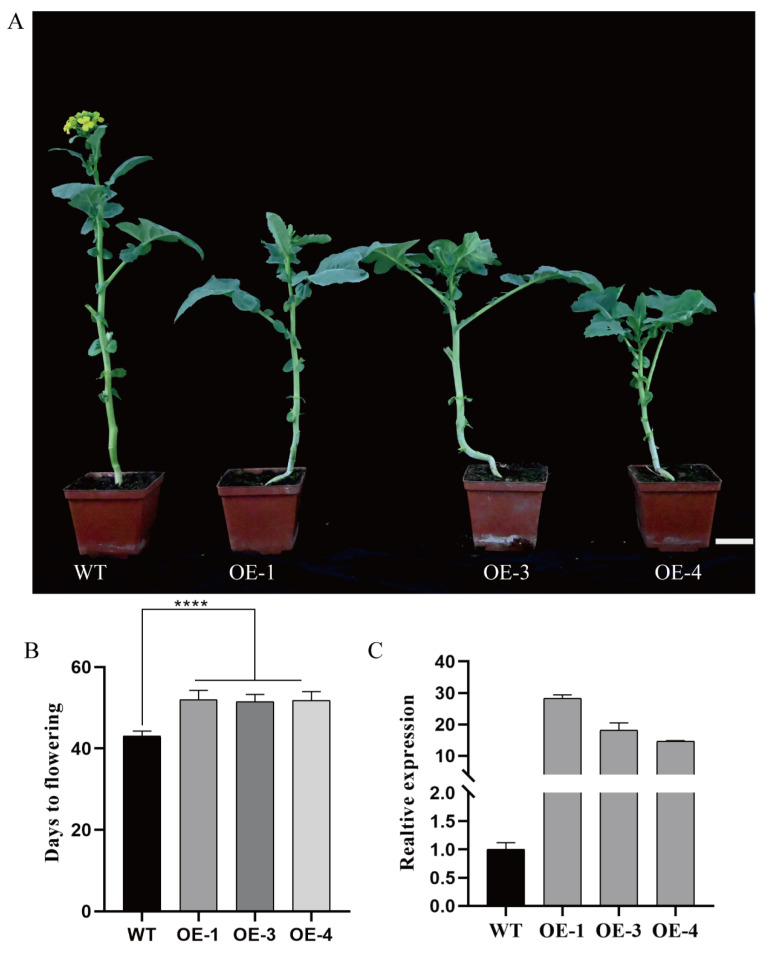
Phenotype of wild-type and *BnaA05COL9* overexpression plants. (**A**) Flowering phenotypes of wild-type and *BnaA05COL9* overexpression plants (OE-1, OE-3, and OE-4; scale bar = 5 cm); (**B**) flowering time statistics of wild-type and *BnaA05*COL9 overexpression plants (**** represents *p*  <  0.0001; Student’s *t*-test, *n*  =  6); (**C**) relative expression level of the *BnaA05* gene in wild-type and overexpression plants.

**Figure 4 ijms-23-14944-f004:**
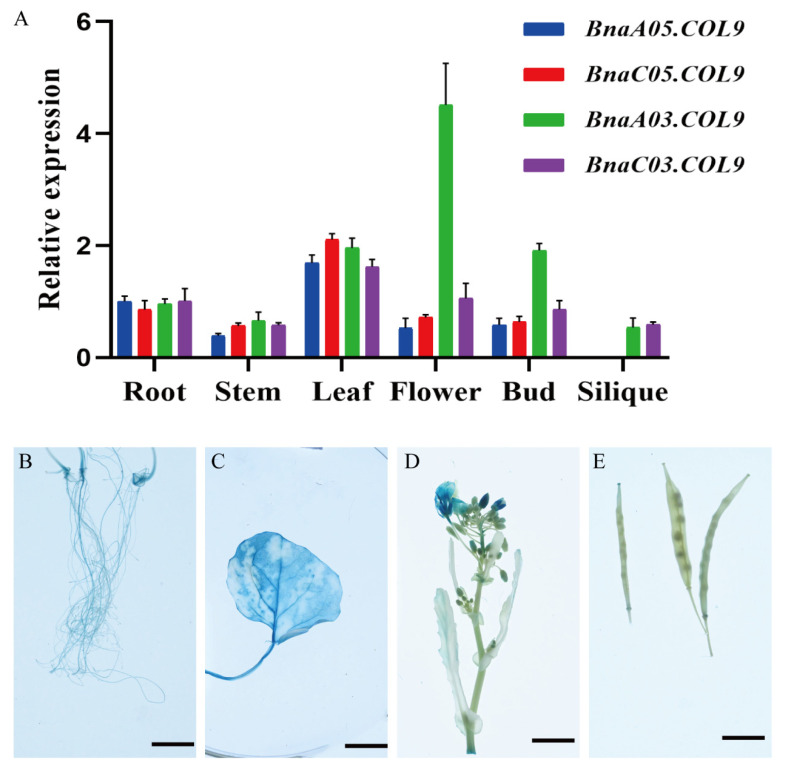
Expression pattern of *BnaCOL9*: (**A**) qRT-PCR analysis of the *BnaCOL9* expression in various tissues, including root, stem, leaf, flower, bud, and silique; (**B**–**E**) GUS staining in *Brassica napus* tissues transformed with the native promoter of *BnaA05COL9* (scale bar = 1.5 cm).

**Figure 5 ijms-23-14944-f005:**
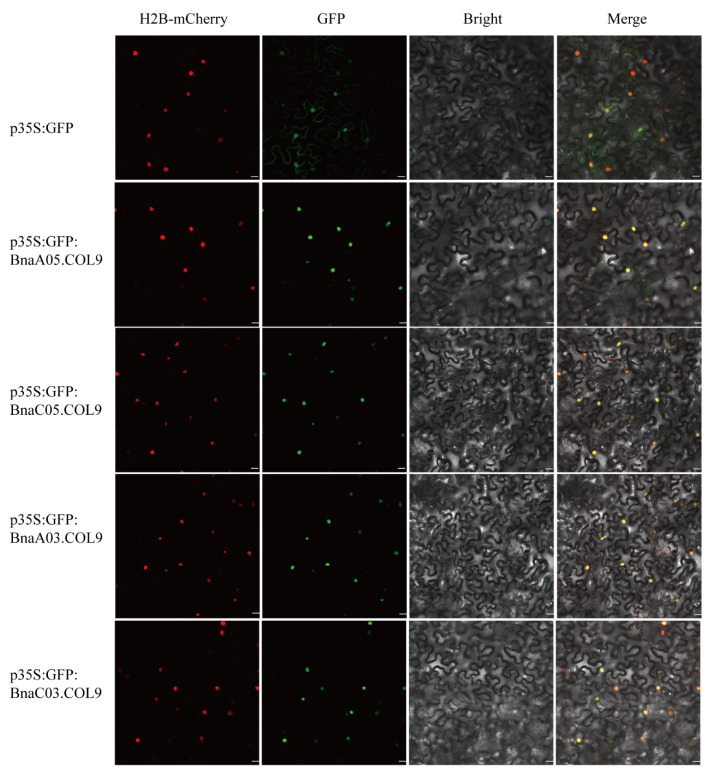
Subcellular localization of *BnaCOL9*: transient expression of a p35S:BnaCOL9-GFP fusion in tobacco (the top row represents the control vector p35S:GFP subcellular location; the lower row represents the p35S: *BnaCOL9*-GFP fusion vector subcellular location. Scale bar = 20 μm).

**Figure 6 ijms-23-14944-f006:**
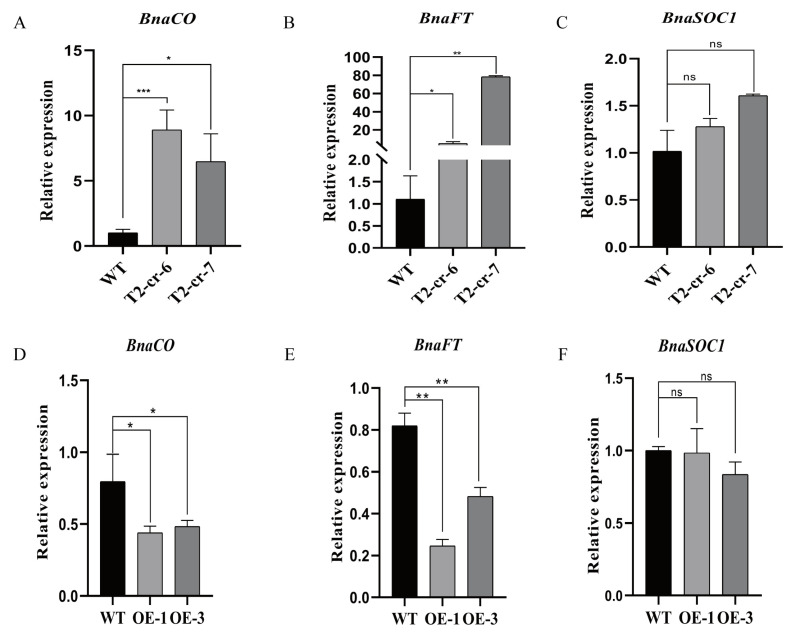
Flowering-related gene expression in transgenic and wild-type plants: (**A**) expression level of *BnaFT* in quadruple mutants; (**B**) expression level of *BnaCO* in quadruple mutants; (**C**) expression level of *BnaSOC1* in quadruple mutants; (**D**) expression level of *BnaFT* in the overexpression lines; (**E**) expression level of *BnaCO* in the overexpression lines; (**F**) expression level of *BnaSOC1* in the overexpression lines. * represents *p* < 0.05; ** represents *p* < 0.01; *** represents *p* < 0.001; ns indicates no significance.

## Data Availability

All data generated or analyzed during this study are included in this published article and its Appendix A.

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
