# Peer review of "CRISPR/Cas9-Mediated Targeted Mutagenesis of BnaCOL9 Advances the Flowering Time of Brassica napus L."

_ijms, 2022, doi:10.3390/ijms232314944_

Round 1

Reviewer 1 Report

The work in general is good and worth publishing as it can be of interest to the community. However, the manuscript lacks several important points without them can not be published.  

Introduction:

It is known that SOC1 (SUPPRESSOR OF OVEREXPRESSION OF CONSTANS), CO (CONSTANS), and FT (FLOWERING LOCUS T) are the main 3 factors controlling flowering in plants.  The author should provide enough explanation in the introduction about what are these gene families and what are their functions, not only giving their abbreviations. The same holds true for the COL9 (CONSTANS-like 9) gene.

Results and methods:

1.       The author did a convincing experiment to prove that the four B. napus orthologs are good candidates for the AtCOL9.

2.       In the legend of Figure 1A please add the triangles of the four orthologs and the circle of At3G07650 to make it easy to understand the figure.

3.       The author showed nicely 3 quadruple mutation lines, however, there is no data at all about transformation efficiency or PCR Cas9 cassette and or gRNAs cassette in these plants.

4.       Please provide how many seeds were tested for transgenicity in T1 generation, and how many were positive or negative for both genes Cas9 and NPTII.

5.       Please provide a PCR image of Cas9 and NPTII for the tested plants.

6.       The author cited a wrong paper for transformation. Please cite the correct paper.

7.       The author adopted the transformation protocol form Arabidopsis dipping, what are the modifications done to transform B. napus?

8.       The author should explain on what bases, they selected these 3 lines and whether there was more line with quadruple mutation or not.

9.       What is the type the sequence shown in Figure 2a? Is it amplicon sequence (NGS) or Sanger sequencing? If it is an amplicon sequence please provide in number how many reads for the shown sequence compared to other reads. If it was sanger sequencing please show the sequencing beaks to indicate whether it was a single or double beak.

10.   Please include the sequence data of T1 generation as well. Do T1 plants have the same deletion or insertion type?

11.   However, the quadruple mutations flowered earlier which means shorter vegetation time, the mutated plants are a lot taller than WT plants. Can the author discuss the reason that mutated plants with shorter vegetation times are much larger?

12.   How do you confirm it is a knock-out mutant? Please provide a structure of a physical map of the gene indicating CDS region and the location of the mutations. As you selected the target motifs in exon 2/3 and 4/5, it can be that the first part of the protein was translated properly, which leads to a partial function. This is important to identify if there is a frameshift mutation has caused a knockout or a knockdown of the gene.

13.   Figure2: indicate in legend B that it is T2 plants

14.   Line 131 replace “overexpression” word with “OE” abbreviation since you used this in the above line. Do this on other occasions as well.

15.   It is very important to explain why the author did not knock out each ortholog separately to dissect the function of each ortholog. The approach followed in this manuscript cannot prove whether one ortholog or more is responsible for the flowering.

16.   Was the expression pattern experiment done on WT plants or in a mutation plant?

17.   Does the GUS gene has an intron? Please provide this information. If the GUS gene does not contain an intron then Agrobacterium can express it and all the signals shown in figure 4 can be from persistent bacterial cells on the plant tissue.

18.   In Figure 6: the shown mutations are cr-1 and cr-2, while in the earlier part of the manuscript like in Figure 2, lines cr-6, 7, and 9 were used for the experiment. Why not the same lines and what is the mutation type of cr-1 and cr-2?

19.   Line 274, the link of the guides designing toll does not work. Please add a working link.

Author Response

Dear Madam/Sir,

Thank you for your valuable suggestions in results section, which will make our article more complete. The manuscript “CRISPR/Cas9-mediated targeted mutagenesis of BnaCOL9 advances flowering time in Brassica napus L.” (ijms-1969141) has been revised with full considerations of the comments and suggestions from you and is now returned for your further reviews. We have responded to your comments point-by-point as follows. We hope that the new manuscript will satisfy you and reach the level of publication.

Specially, we greatly appreciate for your kind and valuable comments to improve our manuscript. Thank you very much.

Best regards,

Sincerely

Bin Yi

Journal: IJMS (ISSN 1422-0067)

Manuscript ID: ijms-1969141

Title: CRISPR/Cas9-mediated targeted mutagenesis of BnaCOL9 advances flowering time in Brassica napus L.

Comments and Suggestions for Authors

The work in general is good and worth publishing as it can be of interest to the community. However, the manuscript lacks several important points without them can not be published. 

Response :Thanks for your kind advice and detailed suggestions. We would like to thank reviewer for giving us constructive suggestions which would help us both in format and in depth to improve the quality of the paper. Here we submit a new version of our manuscript, which has been modified according to the reviewers’ suggestions. We mark all the changes in red in the revised manuscript.

Introduction:

It is known that SOC1 (SUPPRESSOR OF OVEREXPRESSION OF CONSTANS), CO (CONSTANS), and FT (FLOWERING LOCUS T) are the main 3 factors controlling flowering in plants.  The author should provide enough explanation in the introduction about what are these gene families and what are their functions, not only giving their abbreviations. The same holds true for the COL9 (CONSTANS-like 9) gene.

Response: Thanks for your suggestion. We have revised it in the manuscript. Please find it in the revised manuscript.

Results and methods:

  1. The author did a convincing experiment to prove that the four B. napus orthologs are good candidates for the AtCOL9.

Response 1: Thanks for your suggestion. Yes, according to the reference genomic sequence, we designed specific primers for each copy to amplify the coding sequence on the wild type, and by sequencing we found that these four copies are indeed candidates for AtCOL9.

  1. In the legend of Figure 1A please add the triangles of the four orthologs and the circle of At3G07650 to make it easy to understand the figure.

Response 2: Thanks for your kind advice and detailed suggestions. In the legend of Figure 1A, we have added the triangle of four orthologs and the circle of At3G07650.

  1. The author showed nicely 3 quadruple mutation lines, however, there is no data at all about transformation efficiency or PCR Cas9 cassette and or gRNAs cassette in these plants.

Response 3: Thanks for your suggestion. We obtained 40 T0 regenerated transgenic plants and used Cas9-F/R primers for transgene positivity detection and 38 of them were positive. The percentage of positive plants was 95%. As shown in the figure below, 1-40 represent the transgenic plants regenerated by T0, respectively.

  1. Please provide how many seeds were tested for transgenicity in T1 generation, and how many were positive or negative for both genes Cas9 and NPTII.

Response 4: Thanks for your suggestion. Since we tested the regenerated plants positive at T0 generation, The transgenic plants of the T1 generation were all identified positive plants. we only performed editing tests for transgenic plants at T1 generation.

  1. Please provide a PCR image of Cas9 and NPTIIfor the tested plants.

Response 5: Thanks for your suggestion. We detected the PCR images of the test material for cas9 placed below.

  1. The author cited a wrong paper for transformation. Please cite the correct paper.

Response 6: Thanks for your suggestion. We have cited the correct references in the revised manuscript.

  1. The author adopted the transformation protocol form Arabidopsis dipping, what are the modifications done to transform B. napus?

Response 7: Thanks for your suggestion. We performed the genetic transformation of Brassica napus according to the modified method of dai et al [1]. The paper on the experimental method we put below.

  1. Dai, C., Li, Y., Li, L. et al.An efficient Agrobacterium-mediated transformation method using hypocotyl as explants for Brassica napusMol Breeding40, 96 (2020). https://doi.org/10.1007/s11032-020-01174-0
  2. The author should explain on what bases, they selected these 3 lines and whether there was more line with quadruple mutation or not.

Response 8: Thanks for your suggestion. Firstly, we selected these three lines based on the phenotypes of the T1 generation and the results of the editing test. Next, we verified the phenotypes of T1 in T2 generation. We also have other lines with quadruple mutants whose flowering times are similar to those of the quadruple mutations as we showed.

  1. What is the type the sequence shown in Figure 2a? Is it amplicon sequence (NGS) or Sanger sequencing? If it is an amplicon sequence please provide in number how many reads for the shown sequence compared to other reads. If it was sanger sequencing please show the sequencing beaks to indicate whether it was a single or double beak.

Response 9: Thanks for your suggestion. The sequence type in Figure 2A is amplicon sequencing. Three tables were created for the reads sequenced from the amplicons. The table contains their mutant type and the number of reads. For example, in the cr-6 line, the number of reads with A-base insertion in the first target site of the A05 copy was 9815, the number of reads with 4 bases deletion was 4263, and the number of reads without mutations was 933. The number of reads with A-base insertion at the second target site of A05 copy was 24109. In order to better distinguish the reads of each copy, the mutation type for each copy is distinguished by different colors in the table.

Mutant type

Number of sgRNA reads

Mutant type

Number of sgRNA reads

cr-6

A05

+A

9815

-4D(GTTGG)

4263

A

24109

WT

933

C05

+A

10686

+T

14280

WT

2591

+A

1977

+T

4194

+G

2466

A03

+A

10627

+C

25181

WT

2088

+T

1421

+G

280

C03

WT

4905

+T

8331

+A

1540

+A

5575

+T

943

WT

2135

-G

703

-G

1586

cr-7

A05

WT

10529

+A

25926

+A

1758

-4D(GTTG)

1105

C05

-5bp(ACTTG)

5305

WT

26849

+T

3186

+G

2945

WT

1439

A03

WT

4488

+C

25181

+A

3095

+T

2133

+G

1898

C03

WT

4067

+A

28900

+A

3116

+T

2194

-G

2048

-5D(ACTTG)

1058

cr-9

A05

-4D(GTTG)

8777

+A

30760

+A

2853

+T

2152

C05

+A

6675

+A

10872

+T

4897

+T

2526

WT

4212

+G

2809

A03

-G

18630

+C

26441

+A

3808

+T

3066

C03

+A

4970

WT

22428

+T

3460

+A

2403

WT

2383

-G

1966

  1. Please include the sequence data of T1 generation as well. Do T1 plants have the same deletion or insertion type?

Response 10: Thanks for your suggestion. By the editing detection of the T1 generation, the editing of the two lines in the T1 generation was the same except for cr-6 in the A05 copy which was different from the T2 generation. We put the editing situation of the T1 generation in Supplemental Figure 3, please check in the revised version.

  1. However, the quadruple mutations flowered earlier which means shorter vegetation time, the mutated plants are a lot taller than WT plants. Can the author discuss the reason that mutated plants with shorter vegetation times are much larger?

Response 11: Thanks for your suggestion. COL9 gene only affected flowering time of rapeseed, but not plant height. We previously observed that there was no difference in plant height between WT and mutants before bolting in rapeseed. The sharp increase in plant height after flowering was caused by bolting. By reviewing other papers, we found that knock-out of the flowering time related gene BnaSVP could advance the flowering time of rape [1] (as shown in the figure below), which was similar to the phenotype of our col9 mutant.

  1. Ahmar, S.; Zhai, Y.; Huang, H.; Yu, K.; Hafeez Ullah Khan, M.; Shahid, M.; Abdul Samad, R.; Ullah Khan, S.; Amoo, O.; Fan, C. et al. Development of mutants with varying flowering times by targeted editing of multiple SVP gene copies in Brassica napus L. The Crop Journal 2021, doi:10.1016/j.cj.2021.03.023.
  2. How do you confirm it is a knock-out mutant? Please provide a structure of a physical map of the gene indicating CDS region and the location of the mutations. As you selected the target motifs in exon 2/3 and 4/5, it can be that the first part of the protein was translated properly, which leads to a partial function. This is important to identify if there is a frameshift mutation has caused a knockout or a knockdown of the gene.

Response 12: Thanks for your suggestion. As shown in the figure, the black color represents the coding sequence of the gene. Single base insertions and deletions in the coding sequence cause changes in the protein.

  1. Figure2: indicate in legend B that it is T2 plants

Response 13: Thanks for your reminding, we have modified it according to your suggestion.

  1. Line 131 replace “overexpression” word with “OE” abbreviation since you used this in the above line. Do this on other occasions as well.

Response 14: Thanks for your suggestion. We have revised it in the manuscript.

  1. It is very important to explain why the author did not knock out each ortholog separately to dissect the function of each ortholog. The approach followed in this manuscript cannot prove whether one ortholog or more is responsible for the flowering.

 Response 15: Thanks for your suggestion. According to the protein sequence comparison and the analysis of its containing structural domains, we speculated that the four copies of BnaCOL9 may have redundant functions, therefore, we did not design targets for each copy individually to research their functions.

  1. Was the expression pattern experiment done on WT plants or in a mutation plant?

Response 16: Thank you for your comment, expression pattern analysis experiments were done on WT plants.

  1. Does the GUS gene has an intron? Please provide this information. If the GUS gene does not contain an intron, then Agrobacterium can express it and all the signals shown in figure 4 can be from persistent bacterial cells on the plant tissue.

Response 17: Thanks for your reminding, We used a modified GUS vector without introns. GUS gene sequence is presented below.

atgttacgtcctgtagaaaccccaacccgtgaaatcaaaaaactcgacggcctgtgggcattcagtctggatcgcgaaaactgtggaattgatcagcgttggtgggaaagcgcgttacaagaaagccgggcaattgctgtgccaggcagttttaacgatcagttcgccgatgcagatattcgtaattatgcgggcaacgtctggtatcagcgcgaagtctttataccgaaaggttgggcaggccagcgtatcgtgctgcgtttcgatgcggtcactcattacggcaaagtgtgggtcaataatcaggaagtgatggagcatcagggcggctatacgccatttgaagccgatgtcacgccgtatgttattgccgggaaaagtgtacgtatcaccgtttgtgtgaacaacgaactgaactggcagactatcccgccgggaatggtgattaccgacgaaaacggcaagaaaaagcagtcttacttccatgatttctttaactatgccggaatccatcgcagcgtaatgctctacaccacgccgaacacctgggtggacgatatcaccgtggtgacgcatgtcgcgcaagactgtaaccacgcgtctgttgactggcaggtggtggccaatggtgatgtcagcgttgaactgcgtgatgcggatcaacaggtggttgcaactggacaaggcactagcgggactttgcaagtggtgaatccgcacctctggcaaccgggtgaaggttatctctatgaactgtgcgtcacagccaaaagccagacagagtgtgatatctacccgcttcgcgtcggcatccggtcagtggcagtgaagggcgaacagttcctgattaaccacaaaccgttctactttactggctttggtcgtcatgaagatgcggacttgcgtggcaaaggattcgataacgtgctgatggtgcacgaccacgcattaatggactggattggggccaactcctaccgtacctcgcattacccttacgctgaagagatgctcgactgggcagatgaacatggcatcgtggtgattgatgaaactgctgctgtcggctttaacctctctttaggcattggtttcgaagcgggcaacaagccgaaagaactgtacagcgaagaggcagtcaacggggaaactcagcaagcgcacttacaggcgattaaagagctgatagcgcgtgacaaaaaccacccaagcgtggtgatgtggagtattgccaacgaaccggatacccgtccgcaaggtgcacgggaatatttcgcgccactggcggaagcaacgcgtaaactcgacccgacgcgtccgatcacctgcgtcaatgtaatgttctgcgacgctcacaccgataccatcagcgatctctttgatgtgctgtgcctgaaccgttattacggatggtatgtccaaagcggcgatttggaaacggcagagaaggtactggaaaaagaacttctggcctggcaggagaaactgcatcagccgattatcatcaccgaatacggcgtggatacgttagccgggctgcactcaatgtacaccgacatgtggagtgaagagtatcagtgtgcatggctggatatgtatcaccgcgtctttgatcgcgtcagcgccgtcgtcggtgaacaggtatggaatttcgccgattttgcgacctcgcaaggcatattgcgcgttggcggtaacaagaaagggatcttcactcgcgaccgcaaaccgaagtcggcggcttttctgctgcaaaaacgctggactggcatgaacttcggtgaaaaaccgcagcagggaggcaaacaatga

  1. In Figure 6: the shown mutations are cr-1 and cr-2, while in the earlier part of the manuscript like in Figure 2, lines cr-6, 7, and 9 were used for the experiment. Why not the same lines and what is the mutation type of cr-1 and cr-2?

Response 18: Thanks for your suggestion. We used the previous cr-6 and cr-7 lines for the experiments, and we have made modifications in the revised version.

  1. Line 274, the link of the guides designing toll does not work. Please add a working link.

Response 19: Thanks for your reminding. We have added a valid link in the revised version.

Reviewer 2 Report

The study by Dr Yi and colleagues reported the impact of col9 mutations on the flowering time of Rapeseed.  They used CRISPR/Cas9 system to edit four BnaCOL9 homologous genes to generate quadruple mutants with early flowering behavior. The study design was good with conclusive data. I found that the manuscript has the potential to be improved from the present form.

Major comments:

1.     Some parts of the “Material and Methods” section like construction of vectors, and localization study were not present in sufficient.

2.     I may have missed it- please explain why 3rd and 5th exons for BnaA03COL9 were selected as targets unlike other genes.

3.     Did authors study the effect of single, double, and different combinations of triple mutants on flowering time.

4.     What was the rationale behind using two guides?

5.     The discussion section predominantly discussed the previous studies rather than focusing on the study findings. The results and discussion section must be re-written for better understanding/clarity.

6.     Here the authors evaluated transcriptome as their main approach to measure the effect of gene mutations, I recommend including the RNA-Seq studies in the revised version.

7.     Which quadruple mutant line was shown in Figure 2C? Why different sets of data from all the three lines were not shown as this is one of the important figures of the manuscript?

8.     Can Figure 5 be improved?

9.     In Figure 6, data for Cr-1 and Cr-2 lines were presented. Why were cr-6/7/9 not shown?

10.  The nomenclature for over-expressed lines was very confusing. Was it OE-12-X or OE-X?

Minor comments:

1.     The writing must be improved.

2.     Sentence line 303 and 304 were duplicated.

3.     The manuscript missed several references.

4.     Please mention the full form of COL gene once in the manuscript.

Author Response

Dear Madam/Sir,

Thank you for your valuable suggestions in results section, which will make our article more complete. The manuscript “CRISPR/Cas9-mediated targeted mutagenesis of BnaCOL9 advances flowering time in Brassica napus L.” (ijms-1969141) has been revised with full considerations of the comments and suggestions from you and is now returned for your further reviews. We have responded to your comments point-by-point as follows. We hope that the new manuscript will satisfy you and reach the level of publication.

Specially, we greatly appreciate for your kind and valuable comments to improve our manuscript. Thank you very much.

Best regards,

Sincerely

Bin Yi

Journal: IJMS (ISSN 1422-0067)

Manuscript ID: ijms-1969141

Title: CRISPR/Cas9-mediated targeted mutagenesis of BnaCOL9 advances flowering time in Brassica napus L.

Comments and Suggestions for Authors

The study by Dr Yi and colleagues reported the impact of col9 mutations on the flowering time of Rapeseed.  They used CRISPR/Cas9 system to edit four BnaCOL9 homologous genes to generate quadruple mutants with early flowering behavior. The study design was good with conclusive data. I found that the manuscript has the potential to be improved from the present form.

Response:Thanks for your kind advice and detailed suggestions. Those comments are all valuable and very helpful for revising and improving our paper, as well as the important guiding significance to our researches. We have studied comments carefully and have made correction which we hope meet with approval. We have made a major revision to the discussion. We mark all the changes in red in the revised manuscript. The main corrections in the paper and the responds to the reviewer are as flowing:

Major comments:

  1. Some parts of the “Material and Methods” section like construction of vectors, and localization study were not present in sufficient.

Response 1: Thanks for your reminding. We have added the deficiencies in the materials and methods in the revised version.

  1. I may have missed it- please explain why 3rdand 5th exons for BnaA03COL9 were selected as targets unlike other genes.

Response 2: Thanks for your suggestion. To simultaneously knock out four homologous copies of BnaCOL9, we designed two sgRNAs, ensuring that each sgRNA could target to four homologous copies. Therefore, the third and fifth exons were selected for the targeting of BnaA03COL9, respectively.

  1. Did authors study the effect of single, double, and different combinations of triple mutants on flowering time.

Response 3: Thank you for your suggestion, we did not design separate targets for each copy to create single mutations, based on the protein sequence comparison results and gene pattern expression analysis, we presumed that these genes are functionally redundant, and the phenotype of single or double mutations may not be as obvious compared to quadruple mutations, so we only focused on the phenotype of quadruple mutations.

  1. What was the rationale behind using two guides?

Response 4: Thanks for your suggestion. To improve the success rate of being able to target four homologous copies of BnaCOL9 simultaneously, so we designed two sgRNAs with specific sequences.

  1. The discussion section predominantly discussed the previous studies rather than focusing on the study findings. The results and discussion section must be re-written for better understanding/clarity.

Response 5: Thanks for your suggestion. According to your suggestion, we have adjusted the discussion part. The previous studies on the function of BBX gene were deleted from the discussion, and the research results of this topic were mainly discussed. The revised part was marked in red in the manuscript.

  1. Here the authors evaluated transcriptome as their main approach to measure the effect of gene mutations, I recommend including the RNA-Seq studies in the revised version.

Response 6: Thanks for your suggestion. Your suggestion is really good. Through reviewing literatures, we learned that CO, FT and SOC1 are important genes affecting flowering time, and their expression levels in col9 mutants were indeed affected by qRT-PCR. We admit that RNA-seq is a powerful tool to study the gene expression level, we will add this part in the follow-up study and adopt your suggestion.

  1. Which quadruple mutant line was shown in Figure 2C? Why different sets of data from all the three lines were not shown as this is one of the important figures of the manuscript?

Response 7: Thanks for your reminding. We have added the flowering time of the three lines in revised version.

  1. Can Figure 5 be improved?

Response 8: Thanks for your suggestion. We have improved Figure 5, please check it in the revised manuscript.

  1. In Figure 6, data for Cr-1 and Cr-2 lines were presented. Why were cr-6/7/9 not shown?

Response 9: Thank you very much for your reminding. cr-1 and cr-2 represent the cr-6 and cr-7 lines, and we realized that this would be confusing and have modified Figure 6 in the revised version. please check it in the revised manuscript.

  1. The nomenclature for over-expressed lines was very confusing. Was it OE-12-X or OE-X?

Response 10: Thanks for your reminding. We are very sorry that our description confused you and we have made changes to the naming of the over-expressed lines in the manuscript in the revised version.

Minor comments:

  1. The writing must be improved.

Response 1: Thanks for your suggestion. Our manuscript has been touched up by an English editing company.

  1. Sentence line 303 and 304 were duplicated.

Response 2: Thank you very much for your suggestion. We have removed the duplicate parts.

  1. The manuscript missed several references.

Response 3: Thanks for your reminding. We have added to the missing references in the manuscript.

  1. Please mention the full form of COL gene once in the manuscript.

Response 4: Thanks for your suggestion, we have modified the format of COL gene.

Round 2

Reviewer 1 Report

Thanks to author to respond to the addressed questions and comments

General comments:

1.       In general the author did not respond to several questions in a clear way. The author answered on several occasions with wording like “the figure below” but there was no figure below for example in answering point number 11. Such a style makes it difficult to understand the answer. Please specify the figure number in the comments.

2.   The presented work has a major drawback. The quadrable mutant does not give enough evidence of which of the 4 candidate genes is the true ortholog. The protein sequence is not enough. The author should generate single mutation lines by selecting and targeting non-conserved target motifs in each ortholog and analyzing their mutation type, flowering time, and their morphology. Without generating single mutants, it is not possible to know if one of the four candidate genes is not involved in the flowering process of Brassica napus.

Specified comments

1.       Between line 123 and128 the author added a summary of the transformation experiment. The author wrote in he added part the terminology "T0", and this is wrong. In transformation using floral dipping, there is no T0 generation. The T0 terminology is normally used for plants regenerated from tissue culture, not from dipping. The author should change the abbreviation from T to M referring to mutation. For example, the dipped plants are M1 generation and the first harvested seed is M2 generation.

2.       In addition it is not clear what the author meant by 40 T0 plants. Would the author specify which generation exactly is this? Are these 40-dipped plants?  Can the author explain precisely, what he meant by T0 plants on this occasion?

3.       If the author meant by T0 the dipped plants, how the author confirms in the dipped plants that 38 out of 40 plants are Cas9 positive? In the dipped plants, the agrobacterium transfers the T-DNA to the floral part of the plant but not to the vegetative part, hence isolating DNA from the vegetative part of the plant cannot prove whether such a dipped plant is transgenic or not. The first transgenicity test can be done only on the harvested seeds from the dipped plants. Would the author explain this?

4.       Please describe how you screened for the positive plants; did you germinate harvested seeds on a selection medium including antibiotics? Or only did PCR test?

5.       The author claims 95% transformation efficiency, which is far from what is stated in the literature. In floral dipping in Arabidopsis, transformation efficiency is normally around 1% and because of the high production of seeds, such 1% can produce enough seeds.

6.       In lines 165 the subtitle should be in bold

7.       In the point number 9 in my former comments sheet, the author presented a table of the amplicon reads. This table should be included in the supplemental data.

8.       In relation to point number 11 in my former comment sheet, the author did not answer the question of why the quadrable mutation is taller. My question was not whether the COL9 affects the plant height. My question is how does the author explains that the mutated plants that grow for a shorter time are taller?

Author Response

Journal: IJMS (ISSN 1422-0067)

Manuscript ID: ijms-1969141

Title: CRISPR/Cas9-mediated targeted mutagenesis of BnaCOL9 advances flowering time in Brassica napus L.

General comments:

  1. In general the author did not respond to several questions in a clear way. The author answered on several occasions with wording like “the figure below” but there was no figure below for example in answering point number 11. Such a style makes it difficult to understand the answer. Please specify the figure number in the comments.

Response: Thank you for your advice, your suggestions are very helpful to us. We apologize for the unclear wording of our response to the question. We will take care of this issue. Thank you again for your suggestion.

  1. The presented work has a major drawback. The quadrable mutant does not give enough evidence of which of the 4 candidate genes is the true ortholog. The protein sequence is not enough. The author should generate single mutation lines by selecting and targeting non-conserved target motifs in each ortholog and analyzing their mutation type, flowering time, and their morphology. Without generating single mutants, it is not possible to know if one of the four candidate genes is not involved in the flowering process of Brassica napus.

Response: Thanks for your suggestion. Your comments are very helpful to the improvement of our manuscript. Brassica napus is a heterotetraploid species, and each gene exists in at least two homologous copies in Brassica napus. To investigate the function of COL9, we mutated all four homologous copies of COL9 simultaneously to examine its phenotype. We acknowledge that it is not rigorous to construct single and double mutations separately. In the next step, we are also studying the phenotype by isolating single and double mutants from quadruple mutants.

Specified comments

  1. Between line 123 and128 the author added a summary of the transformation experiment. The author wrote in he added part the terminology "T0", and this is wrong. In transformation using floral dipping, there is no T0 generation. The T0 terminology is normally used for plants regenerated from tissue culture, not from dipping. The author should change the abbreviation from T to M referring to mutation. For example, the dipped plants are M1 generation and the first harvested seed is M2 generation.

Response :Thanks for your suggestion. First of all, we apologize for the unclear presentation, our genetic transformation experiment used the hypocotyl of Brassica napus rather than the flower dipping method. The specific transformation procedure was as follows [1].

  1. Seed germination and preparing of hypocotyl explant.

The rapeseed was sterilized with 75% ethanol for 1 minute and then washed 4 times with sterile water. Then, they were sterilized with 0.15% HgCl2 for 15 min and washed 3 times with sterile water. The sterilized seeds were transferred to M0 solid medium in Petri dishes (diameter = 6 cm). and kept in a dark room at 25°C for 7 days. Pick out hypocotyls (carefully remove cotyledons and roots) and cut into pieces of 0.6-0.8 cm in length in Petri dishes (diameter = 9 cm) containing 20 ml of DM medium. Usually, there are 150-200 pieces of explants in one Petri dish.

  1. Preparation of Agrobacterium.

Agrobacterium stain introduced with vector is kept on a solid LB plate with suitable antibiotic. Single colony is picked and put into a glass bottle with 5 mL liquid LB and suitable antibiotic after the seed germination of 5th day. The bottle is fixed in a constant temperature shaker with 200 rpm at 28 °C. Usually, the OD value of the culture solution will reach 0.6–0.8 in 36–48 h.

  1. Infection.

Cultured Agrobacterium is transferred into a 10-mL sterilized tube and centrifuged at 6000 rpm for 10 min. Discard the supernatant, resuspend, and wash the pellet twice by adding 5 mL DM media. Addition of 2 mL suspension solution above into the Petri dish (diameter = 9 cm) which contains prepared explants and 20 mL DM solution. The explants are infected for 30 min with shaking one time every 10 min.

  1. Cocultivation.

Discard the DM solution after finishing the infection step above. Absorb the residual liquid using sterilized filter paper. The explants are transferred onto M1 media for 2–3 days in a dark room at 25 °C.

  1. Callus induction.

The explants are transferred onto M2 media and kept in the tissue culture room for 21 days, at the condition of 16 h day/8 h dark, 26/22 °C.

  1. Shoot differentiation.

The explants are transferred onto M3 media and kept in the tissue culture room for 14 days, at the condition of 16 h day/8 h dark, 26/22 °C. Subsequently, the explants are transferred onto a fresh M3 media every 14 days until the shoots come out.

  1. Root initiation and seedling transplantation.

Green shoot is cut from the junction of the explant, removed the extra dead tissue, and transplanted into the sterilized transparent box containing M4 media. The box is put in the tissue culture room at 26/22 °C under 16 h day/8 h dark. Usually, the root will grow well in half a month.

On the medium of M4, we finally obtained regenerated plants with roots, which we usually named as T0 generation regenerated plants. We transferred these regenerated plants with roots into nutrient soil and placed them in a greenhouse for cultivation.

[1]. Dai, C., Li, Y., Li, L. et al. An efficient Agrobacterium-mediated transformation method using hypocotyl as explants for Brassica napusMol Breeding 40, 96 (2020). https://doi.org/10.1007/s11m032-020-01174-0

  1. In addition it is not clear what the author meant by 40 T0 plants. Would the author specify which generation exactly is this? Are these 40-dipped plants?  Can the author explain precisely, what he meant by T0 plants on this occasion?

Response :Thanks for your suggestion. T0 generation transgenic plants refer to regenerated plants with roots obtained from boxes of M4 medium. We used the hypocotyl of rapeseed for the genetic transformation rather than the flower dipping method.

  1. If the author meant by T0 the dipped plants, how the author confirms in the dipped plants that 38 out of 40 plants are Cas9 positive? In the dipped plants, the agrobacterium transfers the T-DNA to the floral part of the plant but not to the vegetative part, hence isolating DNA from the vegetative part of the plant cannot prove whether such a dipped plant is transgenic or not. The first transgenicity test can be done only on the harvested seeds from the dipped plants. Would the author explain this?

Response :Thanks for your suggestion. Our T0 generation is not a dipped plant. In genetic transformation experiments, we add antibiotics of kanamycin to the medium of M2, M3 and M4. Most of those explants that are not infested with Agrobacterium are screened out. So, we got a total of 40 regenerated transgenic plants. And these transgenic plants were tested by PCR for the presence of Cas9 gene.

  1. Please describe how you screened for the positive plants; did you germinate harvested seeds on a selection medium including antibiotics? Or only did PCR test?

Response :Thanks for your suggestion. In our transgenic experiments, the media targeting the explants all contained the antibiotic kanamycin. Most of those exosomes that were not infested were eliminated. After that, on M3 we would cut off the growth points and put them in the M4 box to culture until these growth points grow roots. Then we transplant the plants with roots into nutrient soil and put them in the greenhouse for cultivation. During this process we will perform PCR on these regenerated plants.

  1. The author claims 95% transformation efficiency, which is far from what is stated in the literature. In floral dipping in Arabidopsis, transformation efficiency is normally around 1% and because of the high production of seeds, such 1% can produce enough seeds.

Response :Thanks for your suggestion. Yes, your point is correct, the experimental conversion with the flower dipping method is indeed not very efficient. But, we used the hypocotyl of rapeseed for the transgenic experiments and added the appropriate antibiotics to the cultures of M2, M3 and M4 for screening. Usually explants that are not infested with Agrobacterium tumefaciens do not undergo the redifferentiation process properly in M3 medium. Such explants will be eliminated.

  1. In lines 165 the subtitle should be in bold

Response :Thanks for your suggestion. We have made changes in the revised version.

  1. In the point number 9 in my former comments sheet, the author presented a table of the amplicon reads. This table should be included in the supplemental data.

 Response :Thanks for your suggestion. We have added the table of amplicon data to the supplemental data. Please check it in the revised version.

  1. In relation to point number 11 in my former comment sheet, the author did not answer the question of why the quadrable mutation is taller. My question was not whether the COL9 affects the plant height. My question is how does the author explains that the mutated plants that grow for a shorter time are taller?

Response :Thank you for your suggestion and we apologize for not explaining this clearly last time. We have grown both mutants and wild type in our greenhouse. Their growth status was the same before the present buds. In the mutant, because the COL9 gene was knocked out. The bud emergence of the mutant was earlier than that of the wild type. After bud emergence, the buds of the mutant will grow and then flower. Because Brassica napus is an infinite inflorescence, its bud platform grows for one stage after flowering, which is why the mutant looks taller than the wild type at a shorter time.

Reviewer 2 Report

After significant revision, this manuscript has improved and would benefit any researcher attempting to do related studies. Thank you to the Authors for addressing my concerns, and I think the manuscript is now more clear and easier to follow. The only concern I have is that the authors should consider including the RNA-Seq data in this manuscript, it would add great value to the manuscript and please add why you did not create the single and double mutants in the manuscript. 

Author Response

Journal: IJMS (ISSN 1422-0067)

Manuscript ID: ijms-1969141

Title: CRISPR/Cas9-mediated targeted mutagenesis of BnaCOL9 advances flowering time in Brassica napus L.

Comments and Suggestions for Authors

After significant revision, this manuscript has improved and would benefit any researcher attempting to do related studies. Thank you to the Authors for addressing my concerns, and I think the manuscript is now more clear and easier to follow. The only concern I have is that the authors should consider including the RNA-Seq data in this manuscript, it would add great value to the manuscript and please add why you did not create the single and double mutants in the manuscript.

Response: Thank you for your suggestions. Your comments have been a great help to the improvement of our manuscript. RNA-Seq is a very useful method to study the regulation of gene expression. We appreciate your suggestions and we will take them on board in our follow-up study. Previous studies have shown that COL9 represses the expression of CO and FT, and we have also confirmed that COL9 represses the expression of CO and FT in Brassica napus by using qRT-PCR in both knockout and overexpression lines. Because Brassica napus is an allotetraploid species, there are at least two copies of each gene. Therefore, we did not aim to create single-process materials. Later, we will separate the corresponding single and double materials through different lines. We explain in the revised version why we did not create single and double mutants. Please check in the revised version.

Round 3

Reviewer 1 Report

Dear Authors,

here are further comments and Suggestions for the Authors

After a couple of rounds of revisions, the manuscript is improved and would be useful for the related community. Thanks to the Authors for being patient and answering the questions. I see the manuscript is now clear and worth publishing. One single concern which is of high importance, the author should run an off-target analysis of the designed gRNAs. this is important to avoid confusing the mutant phenotype. As a hypothesis,  if there is off target, this might explain why the mutated plants are taller than wild-type plants.

Author Response

Journal: IJMS (ISSN 1422-0067)

Manuscript ID: ijms-1969141

Title: CRISPR/Cas9-mediated targeted mutagenesis of BnaCOL9 advances flowering time in Brassica napus L.

Comments and Suggestions for Authors

Dear Authors,

here are further comments and Suggestions for the Authors

After a couple of rounds of revisions, the manuscript is improved and would be useful for the related community. Thanks to the Authors for being patient and answering the questions. I see the manuscript is now clear and worth publishing. One single concern which is of high importance, the author should run an off-target analysis of the designed gRNAs. this is important to avoid confusing the mutant phenotype. As a hypothesis, if there is off target, this might explain why the mutated plants are taller than wild-type plants.

Response: Thank you very much for your suggestions. The questions you raised have helped us a lot in revising the manuscript. We appreciate you taking your valuable time to comment on our manuscript. We used the website CRISPR-P 2.0(http://crispr.hzau.edu.cn/CRISPR2/) to design specific targets against four copies of BnaCOL9. We have chosen targets that are able to specifically identify four copies. Theoretically, the probability of off-target is very low. The hypothesis you raised may exist, and we will follow up with a validation of the off-target. Thank you again for your suggestions, which have helped us tremendously in enhancing our manuscript.